# Fixational saccade inhibition and pupil dilation during self-paced limb movement preparation

**Jan W. Brascamp** *, **Bobicheng Zhang, Vasili Marshev**

Department of Psychology, Michigan State University, East Lansing, Michigan, United States of America

* brascamp@msu.edu

## Abstract

Fixational saccades are modulated in anticipation of several kinds of stimuli and motor actions, suggesting that they can form an overt marker of preparatory state. However, no existing work has studied fixational saccades ahead of spontaneous limb movements in the absence of sensory stimuli, in order to isolate motor preparation from other anticipatory processes (e.g., those related to stimulus processing). Here we examined fixational saccades while participants made self-paced hand and foot movements. We observed that fixational saccade rate steadily dropped prior to either kind of motor action, and recovered immediately after. To examine the relation between this fixational saccade rate signal and other known signals that precede volitional action, we analyzed how this signal related to anticipatory pupil size changes in the same dataset. Replicating previous work, we found steady pupil dilation ahead of limb movements, followed by rapid re-constriction. The amplitude of this pupil signal covaried across individual limb movements with that of the fixational saccade rate signal. The pupil modulations, moreover, followed too shortly after the accompanying fixational saccade rate modulations to be caused by saccade-induced changes in visual input. Together, these observations suggest a joint neural factor influencing both fixational saccade rate and pupil size ahead of limb movements. We discuss possible interpretations of our findings, both specific ones that center on processes of motor planning or temporal expectation, and more general ones that are in terms effort.

## Introduction

Fixational eye movements are involuntary eye movements that occur during intended fixation [1,2]. They come in three types (tremor, drift and fixational saccades), and here our focus is on fixational saccades: ballistic eye movements, generally of small amplitude, that occur during intended fixation. These saccades form a useful marker of internal state. For instance, both the rate and the direction of fixational saccades are modulated while directing spatial attention [3–5], and fixational saccade rate

**Data availability statement:** The data may be found in a public repository at datadryad.org, DOI https://doi.org/10.5061/dryad.mcvdnckbv.

**Funding:** The author(s) received no specific funding for this work.

**Competing interests:** The authors have declared that no competing interests exist.

also briefly drops following the presentation of a sensory stimulus, in a fashion that depends on both stimulus and task particulars [2,6,7].

It is well known that the rate of fixational saccades also reduces in an anticipatory fashion. For example, Betta & Turatto [8] showed saccade inhibition ahead of the predictable appearance of a visual stimulus, more so when the stimulus required a motor response (for related findings see also [9,10]). More recently, Loughnane et al. [11] showed a progressive decrease in fixational saccade rate during decision formation throughout a protracted visual detection interval, leading up to the moment of manual response. Within the domain of eye movements, the rate of fixational saccades has been shown to drop during the preparation of volitional saccades [12–14], even when the latter are generated without any sensory cue.

Such findings suggest a relation between fixational saccades and motor preparation, a relation that may also be considered plausible based on evidence that oculomotor behavior and skeletomotor behavior tend to be coordinated generally [15,16]. Yet, when it comes to fixational saccades, existing findings do not unambiguously point to such a relation with motor preparation. For example, fixational saccade modulations in association with manual responses to sensory stimuli have been tentatively interpreted in terms of motor preparation [8,11], but alongside alternative (or complementary) interpretations in terms of factors like arousal [8] and the collection and accumulation of sensory evidence [9,11,17]. In other words, such results do not isolate any motor-related component of fixational saccade rate modulation, from potential additional components linked to stimulus processing. Observations of fixational saccade modulation ahead of volitional saccade execution, in turn, have invited explanations centered on interactions within the superior colliculus [12,14]. This is reasonable given superior colliculus's central role in oculomotor behavior and established ideas of competitive interactions within this structure [18–20], but this interpretation discourages the idea of a general link between fixational saccade rate and motor preparation, given the superior colliculus's much more limited involvement in skeletomotor action [21].

Here we focus on fixational saccades ahead of self-generated, spontaneous, limb movements (foot and finger presses) in the absence of any cue or sensory-based decision. This means that our paradigm centers on motor actions unrelated to sensory processing and outside of the oculomotor domain. The idea is that any fixational saccade rate patterns observed here, may stem from mechanisms that are related to motor actions generally.

The literature shows several other signals (not involving fixational saccades) that arise ahead of motor actions and thereby betray their imminence, including various electrophysiological signals [11,22–25] as well as similar patterns identified in pupil size records. Regarding the latter: the pupil gradually dilates ahead of spontaneous motor actions, and quickly recovers right after [25–29]. As a step toward placing our findings on fixational saccades in this larger literature, we also measured pupil size changes surrounding spontaneous movements in our task.

Our analysis approach, in general terms, centers on quantifying fixational saccade rate patterns and pupil size patterns during a fixed window time-locked to motor

actions. Aside from quantifying those patterns, we also evaluate covariation between the amplitudes of saccade and pupil patterns across individual motor actions, to examine the possibility of a common origin. Finally, we compare the temporal delay between motor-related saccade and pupil patterns to the temporal delay between visual events and pupil responses (i.e., to the delay of the pupil light reflex), to examine whether some motor-related modulation of pupil size may be caused by visual input changes due to motor-related modulation of fixational saccades.

## Materials and methods

### Participants and task

All experiment procedures were approved by the local Institutional Review Board (name of the institutional review board that approved the study: Michigan State University Institutional Review Board; approval number: STUDY00009635; form of consent obtained: written). 120 participants (average age 24 years, range 18–62 years; 68 male, 51 female, 1 non-binary; all based on self-report) participated in a three-session behavioral experiment involving a battery of tasks, not all analyzed in the context of this project. All participants were enrolled between 1/16/2024 and 4/21/2025. All participants were included in our analyses, but a small number (two for hand movements, eight for foot movements) were excluded from the regression analysis of Fig 5, for reasons explained below. Participants were reimbursed either in course credit or monetarily at a rate of $10 per hour. The first of the three sessions was dedicated to practice on various tasks in the battery, and included about 2 minutes of practice on each of the two tasks relevant to the present study. The remaining two sessions (about 90 minutes each) each involved administration of the full task battery. Those two sessions were each divided up into 15 individual blocks, each for a separate task. Relevant to the present study, one task required participants to make self-paced space bar presses (blocks 4 and 11 of the second session and third session, respectively; block order was fixed for all participants), and one required self-paced foot pedal presses (blocks 12 and 13 of the two sessions, respectively). These tasks' blocks were, in turn, subdivided into runs of 45 seconds each, during which participants executed motor actions at their own pace. Participants were free to rest between runs, and could initiate subsequent runs until 3 minutes into the block, at which point no new runs could be initiated and the experiment proceeded to a subsequent block. This means that, for each effector, participants made their self-paced motor actions during intervals of 45 seconds at a time, and typically (depending on how much rest a participant took between intervals) for about 4 such intervals in succession within a session. This amounts to approximately 5 minutes of data for each of the two effectors across the two sessions. As noted, the majority of task blocks in our battery were not analyzed in the context of this project. For completeness: the non-analyzed tasks, in separate blocks, involved visual or auditory stimuli that were either passively observed or required a discrimination judgment. Although it is likely that oculomotor behavior was impacted by those stimuli and tasks during the blocks during which they were administered, it is unlikely that this systematically influenced our analyses, which center on motor actions made during different task blocks. During the task blocks that involved self-paced motor actions, there was nothing on the screen except a fixation point (a 1 cd/m$^2$ circle with a diameter of 0.15 degrees of visual angle, placed inside a 40 cd/m$^2$ ring with a diameter of 0.3 degrees of visual angle), placed in the center of a 25 cd/m$^2$ background (see Fig 1A for a schematic).

Participants were instructed to use their right foot for the foot pedal, and their dominant hand for the spacebar (87% indicated they were right-handed, 11% left-handed, and 2% ambidextrous, in which case they were free to use either hand). We did not specify which finger or fingers to use for the spacebar. Participants were instructed to direct their gaze at the fixation point while performing these motor actions. Regarding the timing of participants' key/pedal presses, we wished to strike a balance between the actions being spontaneous (i.e., not externally prompted or triggered by explicit counting), yet occurring at a suitable rate (i.e., not so infrequently as to yield little data, nor too frequently to be compatible with our event-related analyses of oculomotor behavior). To invite a suitable rate without encouraging a fixed rhythm or active counting, participants were instructed to press about as frequently as stimulus events occurred during the blocks that were not dedicated to spontaneous motor action. During those blocks (which are not the topic of

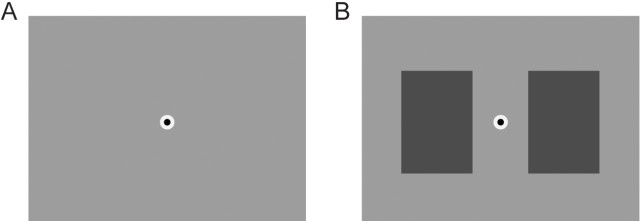

**Fig 1. Stimulus schematic.** A. During task blocks dedicated to spontaneous motor action, only a central fixation point was shown. B. During task blocks during which we measured the pupil light reflex, two rectangles flanked the fixation point and varied in luminance over time.

this study), a range of different stimulus events could happen (see above), always with inter-event intervals selected from an exponential distribution with a scale parameter of 4.6 s and truncated within a 3–8 s range, thereby making for an average inter-event interval of just over 5 s. To encourage compliance with this instruction, participants received an on-screen warning asking to speed up or slow down, respectively, following each run during which they pressed too few or too many times. Such a warning was triggered by comparison to the per-run event numbers during the blocks that were not dedicated to spontaneous motor action: a warning ensued if the number of presses during a spacebar or foot pedal run was not within the central 60% of this distribution of per-run event numbers. Moreover, if a participant received such a warning twice during the same block, this would open a line of communication with the experimenter (see below), who would reiterate the instruction and encourage compliance. To also encourage compliance with the fixation instruction, runs were terminated if the tracker recorded a horizontal gaze position outside of a preset range for over 500 ms consecutively. This preset range was centered on the median gaze position recorded during the first 4 s of a given run, and extended between 2 degrees of visual angle left and right of this median. Note that the use of each run's initial gaze position as a reference here can be considered a drift correction procedure of sorts, yet note that the fixation position itself remained fixed at the center of the screen. Both the choice to only consider horizontal gaze position here, and the choice to allow brief (<500 ms) excursions outside of the range, were meant to prevent blinks from being treated as fixation breaks (with video-based eye trackers, blinks often register as brief vertical deflections in gaze position, owing to the tracker's gaze estimate being affected by the upper eyelid's rapid closing over the pupil and then re-opening). Whenever a run had been terminated like this by a fixation break, the participant would see an on-screen reminder of the fixation instruction prior to the next run. Moreover, if this happened twice during the same block, this would open a line of communication with the experimenter (see below), who would reiterate the fixation instruction and encourage compliance.

One of our analyses involved comparing the temporal delay between the patterns of fixational saccade rate modulation and pupil modulation that accompany motor actions, to the delay between an increase in stimulus luminance and the ensuing pupil light reflex (see Introduction). For this we made use of the pupil light reflex as measured during the same experiments as the remaining data. Specifically, during both the second and the third experiment session, one of the 15 task blocks involved participants passively fixating while a stimulus went back and forth between two luminance levels. The fixation point and screen background were the same as in the blocks with spontaneous motor action (see above), but now there were also two rectangles, positioned left and right from fixation (see Fig 1B for a schematic). Their height and width were 8 and 5 degrees of visual angle, respectively, and their closest side was 1 degree of visual angle from the center of the fixation point. Both rectangles simultaneously alternated between luminance values of 10 cd/m$^2$ and 40 cd/m$^2$, at intervals drawn from the exponential distribution described above. Blocks were subdivided into runs that included between 4 and 12 such alternations (randomly). Otherwise, these blocks were the same as the ones involving spontaneous motor action (with regard to fixation break policy, block duration, etcetera).

Participants performed the experiment inside a darkened room, seated in front of a computer monitor with their head in a head rest. Their eyes were recorded binocularly at a rate of 1000 Hz using an Eyelink 1000 video-based eye tracker (SR Research Ltd.). The eye tracker's pupil size readings had been calibrated using artificial pupils printed on paper [30] to allow pupil diameter to be expressed in real-world units (mm). Gaze location was calibrated once at the start of each session, using Eyelink's 9-point calibration procedure. A researcher monitored experiment progress and the eye tracker's signal quality from a control station next door, and could communicate with the participant through a baby monitor. Motor responses were recorded using a regular USB keyboard and a USB foot pedal (iKKEGOL). These were polled at a rate of 85 Hz, implying that presses were registered at a delay no larger than ~12 ms.

## Eye signal preprocessing

Preprocessing of eye data generally followed the same approach as this lab has used elsewhere [31]. Blinks were identified using a custom algorithm that centered on instances of rapid pupil size change (for this type of eye tracker, the eyelid closing or opening over the pupil causes a rapid change in the registered pupil size) as well as signal absence (no pupil size is recorded while the eyelid covers the pupil). The algorithm also verified that candidate blinks overlapped between both eyes before accepting them as true blinks. Some instances of signal absence were not flagged as a blink by this algorithm (typically because they were not accompanied by fast changes in recorded pupil size and/or because they were monocular); those instances were labeled as missing data rather than blinks. Saccades were detected using a minor variant of Engbert & Mergenthaler's algorithm [32] that identifies saccades by marking periods where gaze displacement velocity deviates strongly from the norm. This algorithm was applied separately to the two eyes, after which any marked time periods (for either eye) that were separated by 5 ms or less were merged into a single candidate saccade. If those time periods came from both eyes' traces, the candidate saccade was labeled as binocular. Candidate saccades that fell within 10 ms from a detected blink were discarded, considering the artifactual gaze displacement that can accompany blinks for video-based eye trackers (see above). We also excluded those saccades that were not labeled as binocular, as well as those with a duration outside of the 8–200 ms range. Pupil size readings were converted to mm and then averaged across eyes, and blinks were interpolated with a linear curve that linked the median pupil size within a window just before the blink to the median size within a window just after. We then computed the difference in pupil size between consecutive samples and low-pass filtered the resulting temporal derivative signal using a third order Butterworth filter with a critical frequency of 25 Hz. We performed further analyses on this pupil size change rate signal.

## Event-related fixational saccade rate analysis

For hand movements and foot movements separately, we computed event-related fixational saccade rates within a time window from 2.5 s before to 3 s after each motor event, for each participant individually. In defining motor events for the foot pedal condition, we treated instances of multiple closely spaced presses (less than 300 ms apart) as a single motor event at the time of the first of those presses. This was because we noticed the foot pedal sometimes registering multiple presses when it was held down (note that this exclusion procedure was used for the visualization of Fig 2, as well). Within the event-related time window, fixational saccade rate was computed per 25-ms time bin, as the total number of saccades tallied in that bin across all hand or foot presses, divided by the product of the bin width in seconds and the total number of hand or foot presses contributing to that tally. In other words, the procedure estimated the fixational saccade rate in saccades per second, at individual time points spaced 25 ms apart around the moment of a hand or foot press. Note that, while saccades could span multiple 25-ms bins, each saccade was assigned to exactly one bin, based on its timestamp which was placed at the midpoint between the saccade's start and end. Both for visualization in Fig 3 and for establishing cluster significance (see section *Statistics*), we smoothed the resulting fixational saccade rate data using a sliding boxcar of 250 ms duration. In cases where two consecutive motor events were so closely spaced that peri-event windows overlapped (extensive occurrence of which was

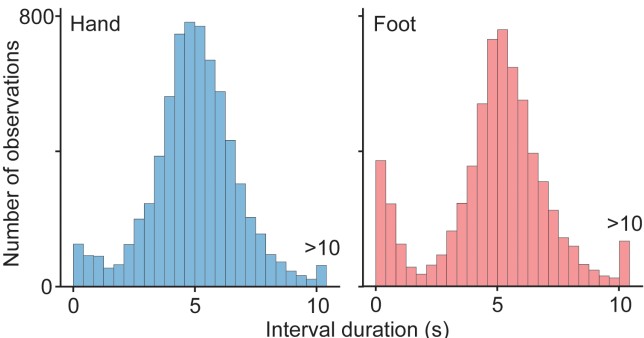

**Fig 2. Intervals between consecutive self-paced motor actions.** Aggregate distribution including all participants, for hand movements (left) and foot movements (right).

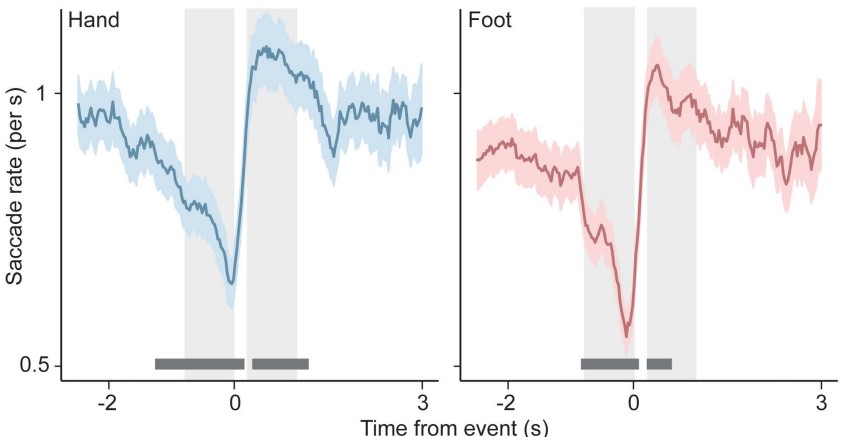

**Fig 3. Rate of fixational saccades surrounding self-paced motor actions.** Fixational saccade rate (across-participant average and standard error) as a function of time relative to hand movements (left) and foot movements (right). Dark bars near the x-axis designate periods of significant deviation from baseline. Light rectangles designate time windows that are used for extracting summary statistics (see panel B and also main text).

discouraged through task instructions; see above), the overlapping part was not included in this rate calculation (i.e., it contributed neither to the saccade count, nor to the denominator used for the rate calculation). This was done to avoid ambiguity as to which of two consecutive motor events may be causally related to the generation or suppression of saccades in a given time period.

Furthermore, after averaging the resulting event-related fixational saccade rate curves across participants (see below), we used visual inspection to define two event-related time windows of interest, both 800 ms in length, during which fixational saccade rates were particularly low and particularly high, respectively. For both hand movements and foot movements, these windows were −0.8 s to 0 s relative to the motor event, and 0.2 s to 1.2 s relative to the motor event. The definition of these two windows allowed metrics of saccade suppression and rebound to be computed for individual motor events, namely the numbers of saccades tallied within the two windows, divided by the windows' widths. The difference between these two metrics was used in the creation of Figs 4 and S1. In this calculation, we ignored all motor events that were closer than 2.5 s to neighboring motor events. Again, the idea was to avoid ambiguity as to which motor event may be causally related to the generation or suppression of saccades during a given time period.

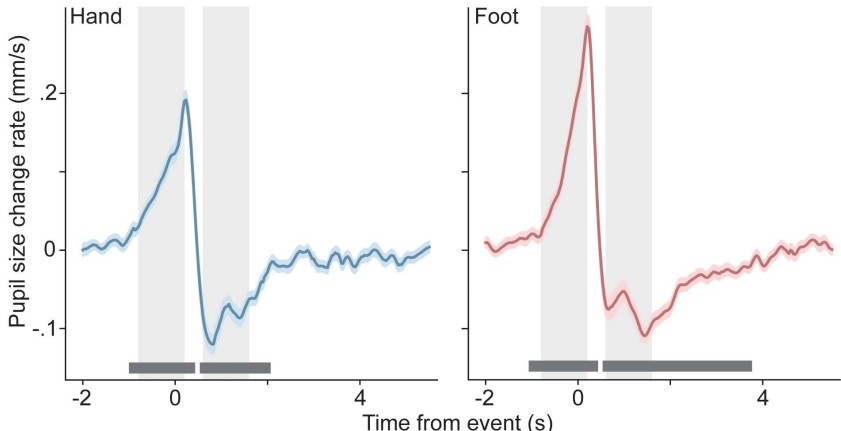

**Fig 4. Pupil size surrounding self-paced motor actions.** Pupil diameter change rate (across-participant average and standard error) as a function of time relative to hand movements (left) and foot movements (right). Dark bars near the x-axis designate periods of significant deviation from 0 mm/s. Light rectangles designate time windows that are used for extracting summary statistics (see Fig 5 and main text).

## Event-related pupil analysis

We computed each participant's event-related pattern of pupil size changes in a time window from 2 s before to 5.5 s after each motor event, using a general linear model approach to deconvolve the event-related pupil signal [31,33,34]. Our motivation for using this approach here, instead of the event-related averaging approach used for fixational saccade rates, is that general linear model approaches deal better with instances where multiple responses overlap. Event-related pupil responses are usually more protracted than fixational saccade rate responses, meaning that there is more such overlap. For example: the constriction elicited by a simple light flash may span from about 400 ms to 1500 ms following the flash [35], whereas fixational saccade rate modulation is usually over within 500 ms following such stimuli [2]. There is evidence that overlapping pupil responses combine approximately linearly [31,34,36,37], while fixational saccade rate responses may not. In the case of such linear combination, general linear model approaches can help disentangle the separate contributions, which event-related averaging cannot. Our specific deconvolution approach was very similar to the one used in [33]: the general linear model used a design matrix that, for each event type, included a staggered set of columns, each with ones at a particular time point relative to the event moments and zeros elsewhere (like a Toeplitz matrix). These time points were 1/30 s apart between the columns in such a set. After solution of the model, these columns' beta weights correspond to the pupil response estimates at those particular time points relative to events of that type. The time series that formed the model's response variable was the temporal derivative signal (pupil size change rate over time) that was computed during pre-processing and then concatenated across the second and third experiment session, but we did integrate some event-related responses back to a size signal again for visualization. The model was solved using ridge regression (using the RidgeCV method of sklearn's linear_model module in Python at default settings).

The movement-related pupil size curves that are reported in the main text, come from a general linear model that included run start moments as an event type, in addition to the button press events themselves. These run start moments were included as nuisance events because they may be followed by pupil responses that overlap with movement-related pupil responses, and their inclusion minimizes the impact of such overlap on our movement-related pupil response estimates. Other work using a general linear model approach to estimating task-related pupil responses [31,33], has included blinks and saccades as additional nuisance events, based on the observation that those events are also typically followed by pupil size changes [33,38] that may overlap with pupil responses of interest. This practice has been questioned, based on the consideration that such oculomotor events may share a common cause with pupil size changes, so that including

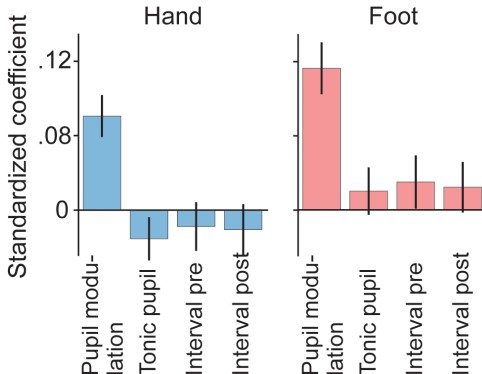

**Fig 5. Association, across individual motor actions within participants, between movement-related modulation of fixational saccade rate and pupil size.** Multiple linear regression results with event-related fixational saccade rate modulation as the response variable. Standardized regression coefficients were computed per participant. 'Interval pre' and 'interval post' correspond to the time periods that separate the current motor action from the previous and subsequent one, respectively. Bars and error bars show means and standard errors across participants, respectively.

the events as nuisance regressors when estimating pupil responses, could amount to regressing out genuine signal rather than noise [39]. To ensure our conclusions' independence of one's choices in this regard, we also estimated movement-related pupil size curves using a model that did include blinks and saccades as additional nuisance variables. As shown in Supporting Information figure S2 Fig, the resulting response curves differed slightly from those shown in the main text, but were overall comparable.

Analogous to our approach for fixational saccade rates, we obtained pupil response metrics for individual motor events by defining two one-second time windows of interest based on visual inspection of the across-participant average of the event-related response. In this case we selected a window from −0.6 s to 0.2 s relative to the motor event as a period of strong pupil dilation, and 0.6 s to 1.6 s relative to the motor event as a period of strong re-constriction (for both hand and foot movements). For each given motor response, average pupil size change rates in those windows served as measures of event-related dilation and re-constriction, respectively. The difference between these two measures was used in the creation of Fig 5. As we did in the case of fixational saccade rates, in this calculation we ignored all motor events that were closer than 2.5 s to neighboring motor events.

## Pupil light reflex

Computation of the pupil light reflex followed the same approach as described for the computation of motor-related pupil signals, except that the general linear model now contained separate regressors for the luminance decrement events and the luminance increment events of the boxes that flanked fixation during the task blocks dedicated to pupil light responses (see above), to estimate the pupil dark reflex and the pupil light reflex, respectively.

## Tonic pupil size

One of our analyses (Fig 5) involved examining whether event-related saccade and pupil responses were associated with tonic pupil size, i.e., whether the magnitude of these transient responses depended on variations in pupil size that occur on a slower timescale. Pupil response magnitude has been reported to exhibit such a dependence in other contexts [38–41]. As our measure of tonic (i.e., slow-varying) pupil size, we started from the preprocessed pupil size signal described above (the signal right before taking the temporal derivative), and slid a boxcar over it with a width of 30 s. To each time point we assigned a tonic pupil size value that was the average within the boxcar centered on that time point, ignoring all missing data as well as data that fell outside of the time point's experiment run.

## Statistics

To identify time windows during which an event-related signal (either fixational saccade rate or pupil size change rate) deviated significantly from baseline at an across-participant level, we used the same cluster-level Monte Carlo analysis [42,43] that we have used previously for this type of data [31]. Taking individual participants' event-related signals as input, this analysis involved first computing, for each timepoint within the event-related analysis window, the two-tailed p-value of a one-sample t-test comparing the across-participant distribution of signal values to the baseline. The next step was to identify clusters of consecutive time points that all had $p < 0.05$ and the same sign of effect. Each cluster was then assigned a 'cluster mass' that was the sum of all these time points' t-values. To examine whether a given cluster mass was more extreme than expected by chance, we compared it to a null distribution obtained by Monte Carlo simulation. Specifically, we performed 1000 iterations of this same cluster mass computation on simulated data: on each iteration, simulated data were created by randomly mirroring or not mirroring each participant's event-related data in the baseline, where 'mirroring' means subtracting the baseline, flipping the sign of the resulting values, and then adding the baseline again. For each iteration, we stored the most extreme cluster mass observed, thus forming a null distribution of 1000 extreme cluster mass values expected by chance, in the absence of systematic signal deviations from baseline. Each cluster mass value from the actual data was then assigned a Monte Carlo p-value equal to the proportion of the null distribution that was more extreme than this value. For the fixational saccade rate signal, the baseline used was the average rate across the first 500 ms and last 1000 ms of the analyzed event-related window. For pupil size change rate, the baseline was 0 mm/s.

To identify whether fixational saccade rate modulation and pupil responses were associated across individual motor actions within participants for a given effector (hand or foot), we assigned each individual motor action a value quantifying fixational saccade rate modulation, and a value quantifying pupil size modulation, respectively. Fixational saccade rate modulation was quantified as the difference in average fixational saccade rate between the post-event rebound window and the pre-event suppression window described above. Pupil size modulation, in turn, was quantified as the difference in pupil size change rate between the pre-event dilation window and the post-event re-constriction window described above. This means that more positive values designated more extensive modulation in both cases. We then assessed the association between fixational saccade rate modulation and pupil size modulation across individual motor actions, by treating the vector of per-event fixational saccade rate modulation magnitudes (with one value per motor action) as the response variable in a multiple linear regression analysis in which other vectors of per-event values served as the independent variables. These independent variables included the vector of per-event pupil response magnitudes, but also vectors corresponding to other variables that could conceivably influence event-related fixational saccade rate modulation and/or pupil size change. These variables were tonic pupil size at the time of each motor action (computed as described above), the time interval separating each motor action from the preceding one, and the time interval separating each motor action from the next. Those latter two variables were included based on the consideration that more closely spaced motor actions might be associated with a different amount of fixational saccade rate modulation and/or pupil size change (either actually, or apparently due to a failure to completely separate modulations associated with closely spaced consecutive motor actions). Both the response variable and the independent variables were z-scored, to make the obtained regression coefficients dimensionless and comparable between independent variables. For each participant, we computed these standardized coefficients as the ones that gave the closest match between the weighted sum of predictor variables and the response variable in a least squares sense (using the lstsq method of numpy's linalg module in Python). For each predictor variable, we assessed significance by comparing the resulting across-participant distribution of coefficient values to 0 using a two-tailed one-sample t-test. For this analysis we excluded two participants from the hand movement condition and eight participants from the foot movement condition, because they had fewer than ten motor events left after excluding motor events that were too closely spaced, as well as motor events that were the first or last of a run (because we could not define the time interval separating those motor events from the preceding/subsequent ones).

To identify, for each effector separately, whether the pattern of pre-event saccade suppression and post-event rebound was consistently observed for individual participants (S1 Fig in the Supporting Information), we simply took the above-described metric that quantifies fixational saccade rate modulation for individual motor actions, and averaged it across all motor actions for a given observer to get an index of that observer's degree of fixational saccade rate modulation.

## Results

We recorded an average of 66 and 62 hand and foot presses, respectively, per participant. Fig 2 shows the distribution of inter-press intervals across all participants for each effector. The figure shows a tendency for participants to sometimes press more frequently than instructed, resulting in an over-representation of very short intervals. (Our analyses were designed to exclude, or mitigate the contribution of, these closely spaced consecutive presses; see Methods.) Never-theless, participants were generally able to comply with the instruction, and produced motor actions approximately every 5 seconds with considerable variability. We recorded an average of 297 saccades per participant per effector condition during the experiment runs (average rate 0.89 saccades per s). Consistent with known kinematic properties of fixational saccades [14,18], median amplitude was 21.3 minutes of arc (interquartile range 17.1 minutes of arc), median duration was 19 ms (interquartile range 14 ms) and the direction of most saccades was close to horizontal (78.5% were angled within 10 degrees from horizontal).

Fig 3 shows the across-participant averaged fixational saccade rates for hand and foot movements, within separate time bins locked to the moment of recorded motor actions. Both plots show a steady decline in fixational saccade rate starting approximately a second ahead of the motor action, and a brief excursion above baseline during the first second following the motor action, followed by a return to baseline. The dark horizontal bars along the bottom of the plots show clusters of time points during which fixational saccade rate differs significantly from baseline (cluster-level two-tailed $p < .05$; see Methods), indicating that both the drop below baseline and the recovery beyond baseline are significant. This pattern of modulation is quite consistent across participants: when comparing fixational saccade rates immediately after movements to fixational saccade rates immediately before movements (i.e., comparing the two time windows marked by the light rectangles in Fig 3), post-movement rates are numerically larger for 85% and 80% of individual participants, for hand and foot movements, respectively (S1 Fig).

The curves of Fig 3 are reminiscent of other signals that surround self-paced movements, for instance a steady pre-movement rise in pupil size, followed by post-movement re-constriction [27–29]. There is evidence that a common neural drive may underlie both saccade generation and pupil size changes in other contexts [39,44]. To assess this possi-bility here, we examined changes in pupil size surrounding motor actions in our experiment, and compared the properties of those changes with those of the observed fixational saccade rate modulation.

Figs 3 shows rate of pupil size change as a function of time relative to hand and foot movements, respectively. For both effectors, the pupils dilate at an increasing rate in the period leading up to the motor action, after which dilation quickly turns into constriction as pupil size returns to baseline. The dark horizontal lines near the x-axes indicate periods where change rate differs significantly from 0 mm/s (cluster-level two-tailed $p < .05$; see Methods). This shows that both the pre-movement dilation and the re-constriction are significant, for both effectors.

There is an obvious resemblance between Figs 3 and 4 (please see S3 Fig in the supporting information for a differ-ent visualization of the pupil data that, instead, prioritizes comparison to existing literature). The main difference is a sign inversion between the two figures, consistent with work showing that saccades, and conditions of high saccade likelihood, are generally associated with pupil constriction rather than dilation [33,39]. It is possible that the resemblance between the saccade and pupil signals in our study reflects a common origin. To evaluate this possibility, we examined the associ-ation between the two signals across motor events within each participant. For each effector, we quantified the extent of fixational saccade rate modulation for each press individually, by computing the fixational saccade rate difference between a time window immediately after, and one immediately before the press (light shaded regions in Fig 3). Similarly, we

subtracted average pupil size change rate between a pre-movement and post-movement window (shaded regions in Fig 4) to get a measure of pupil size modulation for each press individually.

For each participant and effector individually, we then performed a multiple linear regression analysis (see Methods) with event-related fixational saccade rate modulation as the response variable, and event-related pupil size modulation as an independent variable. Three further independent variables were included as potential confounds that may give rise to an across-action association between pupil and saccade response. First, inter-press intervals were variable (Fig 2), and our measures of fixational saccade rate modulation and pupil size modulation for a given motor action may depend on the time interval that separates that motor action from flanking motor actions. Possible reasons for such a dependence include that these signals are truly different for different key press rates, but also that our analysis methods may not fully disentangle signals associated with closely spaced consecutive motor actions (even though they were designed to; see Methods). Therefore, two included confound variables where, for each key press, the duration since the previous press (termed 'interval pre' in Fig 5) and the duration until the next press (termed 'interval post' in Fig 5). Additionally, we considered that event-related pupil responses sometimes correlate (negatively) with slow, or 'tonic', variations in pupil size [33,40]. It is possible that event-related saccade-rate modulation does, too. We therefore included, as a final independent variable, tonic pupil size at the time of the action.

Fig 5 shows the result of this analysis. For both hand presses and foot presses, we found a significantly positive regression coefficient for event-related pupil size modulation (hand: two-tailed $t(117)=4.69$, $p<0.001$; foot: two-tailed $t(111)=5.64$, $p<0.001$), and no significant results for any of the other coefficients (all $p>0.1$). These results indicate that the magnitudes of fixational saccade rate modulation and of pupil size modulation are associated across motor actions, and that this association is not accounted for by variation across motor actions in any of the three confound variables. Note that, in spite of this highly significant evidence for an association when aggregating across participants, the regression coefficient for event-related pupil size modulation did show substantial variability among participants (for hand and foot, respectively, the median coefficients were 0.077 and 0.079, the inter-quartile ranges were 0.28 and 0.26, and the proportions of participants with a value greater than 0 were 0.66 and 0.72).

Finally, we considered the fact that the occurrence of a saccade produces a transient in the visual input that may well elicit a visually-driven pupil response. One might surmise that this, on its own, could cause a relation between fixational saccade rate modulation and pupil size changes in our paradigm: the more saccades, the more visual transients and resulting pupil size changes. We therefore examined whether the temporal relation between the motor-related patterns of fixational saccade rate modulation and pupil size modulation we observed was consistent with this particular kind of causal relation between the two. To this end, we quantified the delay between the across-observer averaged patterns of Figs 2 (fixational saccade rate) and 3 (pupil size change rate), in two ways. First, we computed the latency between the event-related fixational saccade rate minimum just before the motor action and the event-related peak in pupil size change rate that also occurs around that time. This delay was 271 ms for hand movements and 313 ms for foot movements. Second, we computed the delay between the first fixational saccade rate peak just after the motor action and the first minimum in pupil size change rate that happens after the motor action, and found similar values (337 ms and 279 ms for hand and foot movements, respectively). Summarizing, these values indicate that the motor-related pattern of pupil size modulation follows the accompanying fixational saccade rate pattern by a lag on the order of 300 ms (in addition to the patterns having opposite signs). To evaluate whether this lag is consistent with an explanation of the pupil size pattern in terms of saccade-induced visual transients, we measured visually-evoked pupil size modulation in these same participants (see Methods), and quantified the delay between light onset and the moment of the subsequent minimum in pupil size change rate (i.e., the moment of most rapid constriction). Consistent with other estimates [35,45] we found this delay to be 400 ms, so about 100 ms longer than the lag between the motor-related fixational saccade rate signal of Fig 3 and the motor-related pupil signal of Fig 4. (Note that, in the creation of Fig 3, each saccade was timestamped using the midpoint between its start and end moments. Timestamping at the start of the saccade, instead, would not account for the

observed 100 ms difference here, because median saccade duration in our study was only 19 ms.) This argues against saccade-induced visual transients as providing the causal link between those two motor-related signals, instead suggesting an overlap between both signals in terms of neural origin.

## Discussion

We found that spontaneous, self-paced limb movements are associated with systematic changes in the rate of fixational saccades, characterized by a progressive inhibition of fixational saccade rate during the seconds leading up to the movement, and a brief rate increase beyond baseline shortly after. This pattern is reminiscent of fixational saccade inhibition that has been observed surrounding stimulus-related manual responses [8,11] as well as surrounding volitional saccades [12–14]. It is also similar to various electrophysiological [11,22,24,25] and pupil [27–29] signals that accompany limb movements, including self-paced ones. We replicated this earlier pupil finding: a steady rise in pupil size leading up to the motor action, followed by a rapid drop shortly after. Moreover, the extent of the motor-related fixational saccade rate modulation covaried with the magnitude of motor-related pupil size change across individual motor actions. This association could not be accounted for by variation across motor actions in either the relative timing of consecutive motor actions or baseline pupil size. The data also argued against an explanation for the joint occurrence of fixational saccade rate modulation and pupil size modulation in terms of saccade-triggered pupil light responses.

The existing finding that fixational saccades are suppressed in association with stimulus-related (rather than self-paced) motor responses, has prompted various kinds of interpretations, for instance in terms of preparation for predictable stimuli [8], or the sampling and accumulation of sensory evidence [11]. Another possible interpretation that has been offered, is that fixational saccades and limb movements may share common processes of motor preparation and execution, which could lead to response-related saccade modulation [8,11]. By examining spontaneous, self-paced movements, we specifically address accounts such as that latter one, centered on interactions related to motor actions per se, while ruling out accounts centered on external stimuli and their processing. We offer three possible interpretations of the fact that systematic fixational saccade modulation was observed for such self-paced movements as well.

First, consistent with the latter idea articulated above [8,11], motor preparation ahead of self-initiated limb movements may affect, not only skeletomotor actions, but also oculomotor actions. This could cause the observed fixational saccade rate modulation ahead of self-initiated button pressing. Similar modulation, but ahead of self-initiated eye movements, has sometimes been interpreted in terms of inhibitory interactions within the superior colliculus [12,14,46], but our findings invite an interpretation that goes beyond superior colliculus, because this structure's pivotal role does not extend from oculomotor control to skeletomotor control, except in the restricted context of orienting (of head, body, etc.) toward relevant stimuli [21,47]. The present observations, then, are more consistent with other proposals [14] that center on preparatory processes in structures involved in eye and limb movements alike, for example the basal ganglia. Self-initiated limb movements involve activity in an interconnected network that encompasses both the basal ganglia and cortical regions, including pre-movement ramp-up activity that is similar in time course to what is seen in our Figs 2 and 3 [24,48–54]. The basal ganglia also play a key role in saccade suppression and execution [14,20,55–57], raising the possibility that pre-motor buildup activity in the basal ganglia is responsible for graded saccade suppression in our paradigm. This idea is consistent with evidence that oculomotor structures and skeletomotor structures within the basal ganglia are not as cleanly separated as once thought [49,51,56,58–60].

A second perspective on the present observations regarding saccade inhibition places them in the context of temporal expectation. Numerous studies have shown that fixational saccade rates gradually drop ahead of predictable events in general [8,9,13,17], prompting interpretations in terms of temporal expectation and prediction [10,61]. Given that self-generated motor actions are also predictable to the organism, a link with the present findings is conceivable. Interestingly, various pieces of evidence suggest motor system involvement in temporal prediction and also time perception [62–64], raising the possibility that this second perspective on our observations is not completely distinct from the first.

A third possibility is that the interaction, observed here, between limb movements and eye movements, is not specifically related to the motor system, but rather is best viewed from the general perspective of dual task interference. Limb movements require effort, and recent evidence shows that saccades do, too [65–66]. So, perhaps we observe a suppression of fixational saccades surrounding limb movements because both draw on a common processing resource unrelated to motor action per se. This might place the present observations in the same category as findings of fixational saccade suppression associated with, e.g., mental arithmetic [67] or emotional processing [68], collectively suggesting that the rate of (effortful) fixational saccades is reduced whenever effort is allocated elsewhere.

We identified an association, across motor actions, between movement-related fixational saccade rate modulation and movement-related pupil size changes (Fig 5). The observed pattern of pupil size modulation (dilation leading up to the limb movement, and re-constriction shortly after) is consistent with earlier studies showing pupil dilation ahead of motor actions [27–29,69–71]. Several such studies have indicated more pronounced dilation ahead of more demanding motor actions, supporting the idea that this dilation is tied to motor preparation [29,69,71]. We are not aware of prior work showing that these pre-movement pupil dilations are associated on a movement-to-movement basis with concurrent fixational saccade rate signals. The present finding of such an association suggests that the saccade signal and the pupil signal both arise from overlapping neural processes that occur ahead of movement execution. Although our study is not designed to pinpoint the nature of these neural processes, it is reasonable to mention the superior colliculus in this context. Aside from its well-known role in saccade behavior, a variety of evidence shows that the superior colliculus can also contribute to changes in pupil size [72]. For instance, microstimulation of superior colliculus neurons can produce pupil dilations in monkeys [44,73], and pupil size is associated with neural activity in the superior colliculus as measured using single cell recordings [74] or functional imaging [75,76]. If, as speculated above, pre-movement saccade inhibition reflects buildup activity in the basal ganglia, then the causal chain involved plausibly includes the superior colliculus, which receives input originating from the basal ganglia and is centrally involved in saccade production (see above). An account of the accompanying pupil size modulations in terms of activity in that same superior colliculus, therefore, is worth considering. Whatever the neural pathways involved, it is worth repeating that our findings may stem from processes that play a role beyond motor action alone. Fixational saccade inhibition co-occurs with pupil dilation in a range of tasks, typically ones that require effort and including ones that do not involve motor action [39,67,68]. It is possible, therefore, that both pupil size and fixational saccades form an index of effort-related processes that are involved in motor-related tasks and other tasks alike.

In summary, we report fixational saccade rate modulation associated with self-paced hand and foot movements, and we show that this modulation covaries, across movements, with pupil size changes that happen at the same time. These findings are consistent with the idea that motor preparation can give rise to saccade inhibition. Possible interpretations include specific ones that center on pre-movement build-up activity in the basal ganglia affecting both fixational saccades and pupil size via the superior colliculus, as well as general ones that are in terms of non-specific processes of effort allocation.

## Supporting information

**S1 Fig. Saccade rate modulation in individual participants.** To evaluate the consistency across participants of Fig 3's pattern of saccade rate inhibition and rebound, we subtracted, for each individual, average saccade rates between two event-related time windows, indicated by the light shaded regions in the plots of Fig 3, with positive numbers indicating higher saccade rates immediately following motor actions than immediately before. These per-observer difference values are summarized here as strip plots, showing that the majority of participants individually show modulation that is qualitatively consistent with the across-observer average of Fig 3.
(TIF)

**S2 Fig. Pupil diameter change rate (across-participant average and standard error) as a function of time relative to hand movements (left) and foot movements (right), computed in an alternative fashion.** Compared to main text Fig 4, the difference is that the general linear model underlying S2 Fig included oculomor events (saccades and blinks) as nuisance regressors (see Methods). The curves are not identical to those of Fig 4 but do show a very similar pattern, meaning that our results do not importantly depend on this analysis choice. Relatedly, our estimate of the temporal delay between motor-related saccade rate modulation and motor-related pupil size changes (main text, last paragraph before Discussion) remains essentially unaltered when using these curves instead of those of Fig 4 (with these curves, the estimated delay ranges between 238 ms and 313 ms). Dark bars near the x-axis designate periods of significant deviation from 0 mm/s.
(TIF)

**S3 Fig. Pupil size as a function of time relative to hand movements (left) and foot movements (right).** The visualization of pupil data in main text Fig 4 emphasizes the similarity to Fig 3's saccade rate data. To facilitate comparison to existing work, we replot the data of Fig 4 here, this time showing the cumulative sum (Riemann sum) across the values of Fig 4's plots, resulting in curves that denote pupil size relative to the start of the time window (as opposed to pupil size change rate).
(TIF)

## Author contributions

**Conceptualization:** Jan W. Brascamp.

**Data curation:** Jan W. Brascamp.

**Formal analysis:** Jan W. Brascamp.

**Investigation:** Jan W. Brascamp, Bobicheng Zhang, Vasili Marshev.

**Methodology:** Jan W. Brascamp.

**Project administration:** Bobicheng Zhang, Vasili Marshev.

**Software:** Jan W. Brascamp.

**Supervision:** Jan W. Brascamp, Bobicheng Zhang, Vasili Marshev.

**Writing – original draft:** Jan W. Brascamp.

**Writing – review & editing:** Bobicheng Zhang, Vasili Marshev.

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
