## [Decision Letter · Decision Letter 0]

26 Aug 2025

Dear Dr. Brascamp,

Thank you for submitting your manuscript to PLOS ONE. After careful consideration, we feel that it has merit but does not fully meet PLOS ONE’s publication criteria as it currently stands. Therefore, we invite you to submit a revised version of the manuscript that addresses the points raised during the review process.

Both reviewers found the submitted work interesting and suggest a number of changes before re-reviewing it. These comments mainly have to do with the manuscript presentation. In addition, Reviewer 2 proposes an alternative explanation to your results, which should ideally be considered when revising the manuscript. 

We look forward to receiving your revised manuscript.

Kind regards,

Dimitris Voudouris

Academic Editor

PLOS ONE

Journal Requirements:

3. We notice that your supplementary figures are uploaded with the file type 'Figure'. Please amend the file type to 'Supporting Information'. Please ensure that each Supporting Information file has a legend listed in the manuscript after the references list.

Reviewers' comments:

Reviewer's Responses to Questions

**Comments to the Author**

1. Is the manuscript technically sound, and do the data support the conclusions?

Reviewer #1: Yes

Reviewer #2: Yes

2. Has the statistical analysis been performed appropriately and rigorously?

Reviewer #1: No

Reviewer #2: I Don't Know

3. Have the authors made all data underlying the findings in their manuscript fully available?

Reviewer #1: No

Reviewer #2: No

4. Is the manuscript presented in an intelligible fashion and written in standard English?

Reviewer #1: Yes

Reviewer #2: Yes

Reviewer #1: In the submitted manuscript, the authors investigate the eye movement dynamics with respect to hand and foot movements. The novelty of their research is that they used spontaneously generated hand and foot movements independent from the environmental cues or task to uncover the coupling between eye movements and limbic movements in isolation. They showed that the pupil dilates, and the saccade rate drops before the onset of an action, and they recover afterwards. They interpret the correlation, timing, and the relative latency of these eye movements with respect to neurophysiological mechanisms associating motor planning with eye movements.

I believe the authors present interesting research, which offers a unique insight on motor coupling isolated from external signals. However, I think their results are occluded by occasional unclarities and overall detailed repetitions in the current manuscript. Conversely, I believe some key details are missing, although I found the paper already long relative to the number of novel results. I recommend that the authors make a major revision to condense their manuscript by reducing repetitiveness, removing excess details, figures, and analyses, and reconsidering the appropriateness of contents of the individual sections. Below, I added point-by-point comments detailing and specifying these suggestions.

- I believe the introduction and discussion can benefit from introducing Marry Hayhoe’s work on “look-ahead fixations” particularly on hand-eye coordination to introduce the everyday functionality of large saccades with relation to body movements and planning.

- How long/extensive was the training?

- Please clarify L108-L111

- Please report the sampling rate of eyelink used for binocular tracking. Did they calibrate the eye tracker for individual participants? If so, please describe the calibration process and frequency, i.e. every x trials/blocks.

- Where was the fixation point located at the beginning of the experiment? Since the fixation point adaptively repositioned, it would be informative to know the drift from the starting location.

- Please describe the "stimulus events" in Methods. I would particularly appreciate details on the factors that could influence the eye movements investigated (e.g., luminance of different stimuli, inter-stimulus-interval, randomization, etc.). Apart from the description pertaining to the pupil light reflex, the reader has very limited information on the visual stimuli. Relatedly, since there were different "events" occurring on the screen, I wonder if participants had similar baseline eye movements for different events.

- The experiment seems to impose a very strict control over the participants with visual warnings and dialogs with the experimenters. I wonder if and how these warnings and dialogs influence the spontaneity of subsequent actions. These warnings by themselves can be considered as "events" that alter participants’ oculomotor response. Did the authors check those instances for systematic effects?

- I wonder how the authors made sure that the usb inputs and eye tracker data collected by different devices and software with different latencies are aligned.

- I would like to thank the authors for using pupil area change in addition to the loss signal to separate actual blinks from signal loss. If they also checked if blinks were binocular as a criterion, they should mention it.

- To improve the flow of the paper, I strongly recommend authors provide motivation for their analyses and measurements (e.g. pupil light reflex) in the introduction rather than promising it in the methods for the result section.

- How did the authors rationalize their selection of time window and box car of saccade rate analysis? What was the typical range of fixational saccade duration?

- If I understood correctly the authors counted the occurrence of saccades separately for 25 ms bins within the ~5 second time window. I really think that this section would benefit from clearer description. More importantly, I am curious to hear if and how saccades spanning multiple bins are handled. Finally, if the authors believe this is a better approach than simply calculating the saccade rate for a larger time window before/after action and compare it to baseline calculated for the same time window at a different point in the experiments.

- L278 Typo "due"

- L315 Please explain what is meant by "mirroring".

- The authors calculate the difference of differences for checking the correlation between pupil size and saccade rate changes before and after an action per effector. As they mentioned, with this type of data there can be various limitations for statistical analysis. Since their data is unlikely to be normally distributed the authors should use a non-parametric test like Spearman's rho or provide normality tests. An alternative would be to calculate the slope instead of correlation, which would give direction and strength.

- I am genuinely puzzled as to why authors did the correlational analyses if they have thought of potential confounding factors. For clarity and brevity, I strongly recommend them to use one statistical analysis per research question. For the multilinear regression selection of pupil size or saccade rate as the response seems to be arbitrary. If the authors want to account for the tonic pupil size in the saccade-pupil relationship, they could do a partial correlation to remove the variance common to tonic pupil size. However, I believe they should keep one of the analyses.

- L386 please give saccade/sec for comparability with the wider literature.

- I believe the consideration of individual differences in the results section should be mentioned in the discussion. The related results in Figure 2B could be moved to the supplementary materials for brevity, as the authors neither conduct further analysis based on it nor use it for the interpretation of the results.

- In my opinion, the comparison of results with other studies (L417 onwards) are better kept for discussion to preserve the flow of the study findings.

- Again, I believe re-plotting the data for the sake of comparability in the result section distracts the reader and it would be better appreciated in supplementary.

Reviewer #2: The authors describe results of a study investigating microsaccade rates and pupillary changes surrounding self-initiated limb movements (arm, foot). They find a signficiant modulation of microsaccade rate and pupil size change, consistent with earlier works. Interestingly, both microsaccade rate and pupil dilation were linked: The more pupil dilation, the stronger the saccadic suppression. While the differing intervals for the motor responses are a bit odd at first, this nicely excludes concurrent effects of external cues on both pupil size and microsaccade rates.I found the manuscript interesting to read. I have a number of points that may help streamline manuscript and visualization though as the manuscript, in present shape, is less straight-fowrard than necessary. Most importantly, I believe that a third (better?) possibility should be considered as explanation to the authors’ findings. This may be at odds with a mere motor inference interpretation.

In order of points in the manuscript:

Can the authors add a clear paradigm figure showing screens etc? Right now the reader has to infer a lot from the text with room for misinterpretations. I was not particularly concerned of a visual event causing the effects (as the authors also expressed in their control analysis), but a visualization of the screen would be helpful to that end.

P7., ; 139 following: I must admit I found the central gaze check rather odd. 500ms are long, also for blinks. It is unclear to me why the gaze check is sensible to conduct only in the horizontal plane then, as regular blinks should not take substantially longer. Even if so, one may question the use of such data as the intervals interpolated become excessive? Did the authors calibrate the pupil to millimeters alone or also apply the PFE calibration described in the paper by Hayes and Petrov? If they did not do the latter, it might be sensible to add a control analyses (perhaps in a supplement) that uses gaze position as a covariate.

P11., l222: This may benefit from some general information on response latencies of pupil size. Bergamin & Kardon, 2003 may provide such numbers. In our own data, we usually observe times around 300-350ms. Similarly, response latencies for externally triggered saccades may be of interest, so that both timings can be compared.

P.13 From a reader’s perspective, the section on the pupil light reflex comes totally out of the blue. I personally don’t think it flows well to cross reference to a later section for motivation. Please motivate clearly how this is needed. If needed, results should be accompanied by a figure ideally.

P. 18. The first two sentences of the results are repeating parts of the methods, some oddly specific, some very general, I would remove these. Generally, there are many cross references (to methods in the results, to results in the methods) that should be avoided in my opinion.

Motor responses are indeed very reliably found to affect pupil size, Richer & Beatty is a good reference for that. However, similar effects have been described also substantially earlier by Bumke among others (1911, cf Loewenfeld 1999, Strauch 2024, TINS) which might be worth noting.

The authors sometimes call microsaccade rate ‘saccade rate’ If I’m not mistaken. I would advise to use ‘microsaccade rate’ consistently if that’s what they calculated (explicitly discarding saccades I believe), see Figure 2, caption.

P20. L410 to 415 can be cut in my opinion. The authors do not need to describe how a boxplot works. Ideally, the authors should convert this figure to a stripplot that displays all data points.

I wondered about the statistics reported on p23. While the effects are clear, it seems that they underuse the data. Wouldn’t it be better to perform a linear mixed effects model the predicts the change in saccade rate (or number of saccades per interval using an ordinal linear mixed effects model) from pupil size and the other predictors? Or vice versa pupil size from microsaccade rates and other predictors? That way, the authors could also simply take deviations in gaze position, blinks etc into account per trial.

For Figure 4, I think it would make the manuscript a lot nicer to plot faint regression lines per participant (i.e., within a participants’ trials) together with an overall regression line. Of course there will be substantial variation, but this may help assessing the robustness and consistency of the effects, akin to the proposed stripplot.

The authors, in the discussion, speculate about candidate regions in the brain that may underlie the observed link. Personally, I think that this could be a bit more concise, given that the present study allows only limited inference on brain regions (causally) involved beyond primary motor cortex, LC, SC perhaps.

Lastly and perhaps most importantly, I would like to present a third explanation to the authors’ findings. Perhaps it is not that motor actions specifically that interfere with (microsaccadic) eye movements. There is a broad literature demonstrating a perhaps surprising more general link between pupil dilation and (micro)saccade rates. E.g., as the authors will know, pupils dilate with any increase in effort, be it mental or physical (e.g. Bumke 1911, Hess & Polt, 1964, Mathot 2013, Strauch et al., 2022, etc.). Similarly, microsaccade rates drop under increased mental effort (https://doi.org/10.1111/ejn.12395), by now an almost as well replicated finding as for pupil dilations. I therefore consider a motor-specific explanation as quite unlikely – why would this link then also extend to mental effort (see also for another example https://doi.org/10.1167/jov.25.4.16)? These results may be (more) parsimoniously unified however, if one looks at them from a processing bottleneck or dual-task perspective. Assuming that any (including eye) movement is effortful due to its computation and execution costs, results are to be expected. For instance, we showed that saccades, just as larger scale movements, are linked to movement-specific pupil dilations indicating that they require differing degrees of effort (https://doi.org/10.7554/eLife.97760.3 , https://doi.org/10.1177/09567976231179378). These effort-signatures in turn are quite indicative of behavior, i.e., humans make cheaper eye movements when given the choice (https://doi.org/10.7554/eLife.97760.3 , see also https://doi.org/10.1101/2022.06.03.494508). If, in line with the previously mentioned papers, all movements, arm, leg, eye are considered as effortful, the here demonstrated results make perfect sense: when one movement (or mental operation) requires more computational resources, likely inferable via LC activity and thus pupil size change, then they will interfere with each other (as the present manuscript shows). I personally consider this a more likely explanation to the authors’ results than the present interpretations (although they go a little bit in this direction too) and believe that the paper might make an additional valuable contribution to that literature (but I’m biased here, of course).

I could not open the data repository upon clicking on the doi, so could not check data or scripts.

Signed,

Christoph Strauch

Utrecht University

**Do you want your identity to be public for this peer review?** For information about this choice, including consent withdrawal, please see our Privacy Policy

Reviewer #1: No

Reviewer #2: **Yes: ** Christoph Strauch

---

## [Author Response · Author response to Decision Letter 1]

18 Sep 2025

Please see the Response to Reviewers document for detailed responses to the comments raised.

---

## [Decision Letter · Decision Letter 1]

6 Oct 2025

Dear Dr. Brascamp,

We look forward to receiving your revised manuscript.

Kind regards,

Dimitris Voudouris

Academic Editor

PLOS ONE

Journal Requirements:

Reviewers' comments:

Reviewer's Responses to Questions

**Comments to the Author**

Reviewer #1: All comments have been addressed

Reviewer #2: All comments have been addressed

2. Is the manuscript technically sound, and do the data support the conclusions?

Reviewer #1: Yes

Reviewer #2: Yes

3. Has the statistical analysis been performed appropriately and rigorously?

Reviewer #1: Yes

Reviewer #2: Yes

4. Have the authors made all data underlying the findings in their manuscript fully available?

Reviewer #1: No

Reviewer #2: No

5. Is the manuscript presented in an intelligible fashion and written in standard English?

Reviewer #1: Yes

Reviewer #2: Yes

Reviewer #1: The authors performed a major revision addressing my comments on clarity/brevity and reservations about statistical tests and experimental methods. I believe the current manuscript states its purpose and methods in a clear way to interpret the results and discussion. Therefore, I thank the authors for their thorough revision and recommend the paper for publication. Finally, I would like to point that I could still not access the raw data from the provided doi.

Reviewer #2: The authors have addressed my points.

I was still unable to open the data link though - please double check.

**Do you want your identity to be public for this peer review?** For information about this choice, including consent withdrawal, please see our Privacy Policy

Reviewer #1: **Yes: ** Nedim Goktepe

Reviewer #2: **Yes: ** Christoph Strauch

---

## [Author Response · Author response to Decision Letter 2]

9 Oct 2025

We thank the reviewers for their work to improve our manuscript.

---

## [Editor Report · Decision Letter 2]

14 Oct 2025

Fixational saccade inhibition and pupil dilation during self-paced limb movement preparation

PONE-D-25-24924R2

Dear Dr. Brascamp,

We’re pleased to inform you that your manuscript has been judged scientifically suitable for publication and will be formally accepted for publication once it meets all outstanding technical requirements.

Kind regards,

Dimitris Voudouris

Academic Editor

PLOS ONE
---

## [Editor Report · Acceptance letter]

PONE-D-25-24924R2

PLOS ONE

Dear Dr. Brascamp,

I'm pleased to inform you that your manuscript has been deemed suitable for publication in PLOS ONE. Congratulations! Your manuscript is now being handed over to our production team.

Kind regards,

on behalf of

Dr. Dimitris Voudouris

Academic Editor

PLOS ONE